# Pasteurella Multocida Infection in Humans

**DOI:** 10.3390/pathogens12101210

**Published:** 2023-10-01

**Authors:** Marcin Piorunek, Beata Brajer-Luftmann, Jarosław Walkowiak

**Affiliations:** 1Veterinary Practice Marcin Piorunek, 60-185 Skórzewo, Poland; 2Department of Pulmonology, Allergology and Pulmonary Oncology, Poznań University of Medical Sciences, 60-569 Poznań, Poland; bbrajer@ump.edu.pl; 3Department of Pediatric Gastroenterology and Metabolic Diseases, Poznań University of Medical Sciences, 60-572 Poznań, Poland; jwalkowiak@ump.edu.pl

**Keywords:** pasteurellosis, zoonosis, pets, infection transmission, clinical manifestations, complications, treatment

## Abstract

*Pasteurella multocida* (*P. multocida*) is an immobile, anaerobic, Gram-negative coccobacillus fermenting bacterium. This pathogen is commonly prevalent in the upper airways of healthy pets, such as cats and dogs, but was also confirmed in domestic cattle, rabbits, pigs, birds, and various wild animals. Infection in humans occurs as a result of biting, scratching, or licking by animals and contact with nasopharyngeal secretions. Inflammation at the site of infection develops within the first day from the injury. It is usually confined to the skin and subcutaneous tissue but, in particular situations, may spread to other organs and manifest as a severe systemic infection. Careful history-taking and microbiological confirmation of the infection enable diagnosis and appropriate treatment. Any wound resulting from an animal bite should be disinfected. The preferred and highly effective treatment against local *P. multocida* infection is penicillin or its derivatives. The prognosis for *P. multocida* infections depends on the infected site and the patient’s comorbidities.

## 1. Introduction

*Pasteurella multocida* (*P. multocida*) is an immobile, anaerobic, Gram-negative coccobacillus fermenting bacterium belonging to the *Pasteurellaceae* family [1]. This bacterium was first isolated by Kitt in 1878 during an epidemic disease among wild hogs and two years later, described by Pasteur as fowl cholera [2]. In his honor, the bacterium was named *Pasteurella* [3]. *P. multocida* has been serologically classified into five commonly isolated serogroups (A, B, D, E, and F) according to the composition of the polysaccharide capsule and 16 lipopolysaccharides (LPS) serovars using Carter’s and Heddleston’s methods [4,5,6,7]. It is known that the subtypes of *P. multocida* differ in their virulence. Most infections are caused by serogroups A and D, and the most important contributors to bacterial virulence are the polysaccharide capsule, the surface lipopolysaccharide molecule, iron acquisition proteins, and *P. multocida* toxin [8]. Recently, attempts have been made to assess the virulence of this pathogen using genetic methods. The availability of *P. multocida* genome sequences makes it possible to explain the underlying genetic mechanisms of *P. multocida* fitness and virulence using whole-genome sequences, genotypes, including the capsular genotypes, lipopolysaccharide (LPS) genotypes, and multilocus sequence typing, as well as virulence factor-encoding genes of *P. multocida* isolates [9,10].

*P. multocida* is commonly prevalent in various parts of the upper airways of healthy pets, most often cats and dogs, but it was also confirmed in domestic cattle, rabbits, pigs, birds, and wild animals. On the other hand, *P. multocida* may be associated with numerous serious, acute, or chronic diseases in a wide range of animal species, including fowl cholera, bovine hemorrhagic septicemia, enzootic pneumonia of cattle, porcine atrophic rhinitis, and pasteurellosis in pigs [11]. The infection is transmitted to humans most commonly by cats and dogs, with carrier rates of 70% to 90% and 20% to 50%, respectively [1]. In 2021, purebred cats and mixed breeds were the most frequently chosen pet in the European Union (EU), and their number that year was estimated at 113 million [12]. Poland ranked 6th with an estimated 4.082 million cats [13]. The number of domestic dogs in the EU in 2021 has been estimated at over 92 million, and in Poland at 6.466 million [14]. A questionnaire study on the characteristics of the population of dogs and cats in Poland showed that 50% of the inhabitants are dog owners. The percentage of people who own cats is smaller but also significant and amounts to 37% [15].

## 2. The Pathogenesis and Epidemiology of *P. multocida* Infection

Transmission of infection from animal to human occurs through biting, scratching, or licking. Four to five million animal bite wounds are reported annually in the USA, resulting in approximately 300,000 emergency department visits [16]. The bacteria are isolated from approx. 50% of wounds caused by dogs and 75% of wounds caused by cats [8]. Minor scratches and bites caused by pets are not reported by the healthcare facility, which is just one reason why it is difficult to determine the prevalence of *P. multocida* infection in humans. Although the number of bites caused by dogs is clearly higher than those caused by cats, only 3–18% of all dog bites result in a wound infection, compared to 20–80% of cat bite infections. It is also possible to become infected without direct human–animal contact, such as contamination of a wound with dog saliva or through socks covered with cat hair and dandruff, as well as by eating food that the animal previously ate. Infection of this type is associated with the presence of comorbidities, and in the worst case, may be life-threatening [17]. There are known single cases of wound infections in humans resulting from bites by tigers, leopards, lynxes, rats, opossums, lions, horses, rabbits, boars, and panthers [18]. Five different species or subspecies cause pasteurellosis in humans, including *P. multocida*, *P. septica*, *P. canis*, *P. stomatis*, and *P. dagmatis*. Isolation and identification of the pathogen are not particularly difficult [19]. It is estimated that *P. multocida* is the most commonly cultured bacterium from infected human wounds [20].

## 3. Diagnosis of *P. multocida* in Human

### 3.1. Clinical Suspicion and Evaluation

*Pasteurella* spp. should first be considered as a cause of soft tissue infection resulting from scratching, biting, or licking by a cat or dog. The infection is characterized by a rapid onset and intense inflammatory reaction, the course of which is similar to soft tissue infection caused by group A streptococci, such as *Streptococcus pyogenes* [21,22,23]. Cat bite wounds can penetrate deep into the soft tissues, which is related to the structure of their teeth. Compared to dog bites, this poses a higher risk of osteomyelitis, tenosynovitis, and septic arthritis, as well as septic arthritis of damaged joints and joint prostheses [24]. An important factor for *P. multocida* infection that should be considered are patients with compromised immunity, liver failure, diabetes, malignant tumors, and severe infections such as pneumonia, meningitis, or bacterial peritonitis who have been exposed to animals [1].

### 3.2. Microbiological Examination

Diagnosis of *P. multocida* infection in diagnostic material is possible by isolating the pathogen from a culture, using PCR or serological tests. Due to the antigenic complexity of the bacteria and the undetermined taxonomic status of many *Pasteurellaceae*, serological methods do not enable the diagnosis of an active infection and are more likely to produce false-positive results compared to viral serology. The biological material for a culture may be a wound swab, sputum, blood, cerebrospinal fluid, synovial fluid, and others, depending on the location of the infection [8].

The recommended medium for growing *P. multocida* is 5% Columbia Agar with sheep’s blood, maintained at a temperature of 37 degrees Celsius for 24–72 h, in an atmosphere of 10% CO2 [25]. Pathogen identification is performed using some methods, such as Vitek2, VITEK MS, Bruker Biotyper MS tests, and traditional biochemical tests [26]. Drug susceptibility is performed according to the EUCAST standard [27]. A fast, sensitive, specific, and highly effective method for identifying *P. multocida* and its subspecies is the PCR test. It is an alternative to biochemical methods (fermentation of sorbitol and dulcitol) and others [28].

## 4. Host-Pathogen Interactions of *P. multocida* and Clinical Manifestation in Humans

The infection in humans usually has a rapid course. Limited superficial soft tissue infection and abscesses are most commonly reported in healthy individuals. Still, in particular situations, such as immunodeficiencies, *P. multocida* may spread to other organs and manifest as a severe systemic infection [29]. Inflammation at the site of infection develops within 24 h of injury and is most often confined to the skin and subcutaneous tissue. Bites and scratches caused by a cat usually affect the upper limbs and face. Their narrow, sharp teeth cause a small crack in the skin, which quickly closes up, causing the entrapment of bacteria in deeper tissues [30]. The typical clinical manifestations associated with inflammation include wound-related erythema, tenderness at the wound site, edema, purulent exudate, and cellulitis [31,32,33,34]. As a result of contact with animal secretions, infection of the lower respiratory tract and the development of pneumonia, tracheitis and bronchitis occur relatively often, as well as serious complications such as lung abscess or pleural empyema are possible. The risk factors are old age, chronic respiratory diseases such as COPD or bronchiectasis, and malignant tumors [35]. The prevalence of respiratory tract infections caused by *P. multocida* in everyday medical practice has not been determined [36]. In a review of 108 patients with *P. multocida* infection of the respiratory system, 49 cases of pneumonia, 37 of tracheobronchitis, 25 of pleural empyema, and 3 cases of lung abscess were reported [37]. Another study [31] analyzed 136 *P. multocida* infections non-related to animal bites and found that 80 were related to the respiratory tract of people with chronic lung disease. *P. multocida* may be a commensal organism in the respiratory system of patients with underlying lung disease. The infection may be mild, and the isolation of the pathogen from the respiratory tract may be random and difficult to interpret [38].

Upper airway pasteurellosis is exceptional and is secondary to oropharyngeal carriage or contamination of the patient’s airways by domestic or farm animals. The transmission mode may be inhalation of infectious nasopharyngeal secretions from cats and dogs. In the available literature, only one fatal case of *P. multocida* epiglottitis infection with associated septicemia was found [39,40,41].

Fatal *P. multocida* septicemia and necrotizing fasciitis in patients with gouty arthritis and open wound with tophi licked by an infected domestic dog has also been described [22]. Similarly, prosthetic joint infections (PJI) have rarely been reported in science publications. PJI typically occurs in the context of an animal licking the wound, bite, or scratch. They may be related to perioperative complications of hip or knee arthroplasty, such as hematoma formation, superficial wound infection, wound drainage, and wound dehiscence. Older age, diabetes, rheumatoid arthritis, immunosuppressive therapy, malignancy, and arthroplasty revision have been identified as risk factors [42,43,44,45,46]. A single case of pubic symphysis septic arthritis caused by *P. multocida* has been described too [47].

Meningitis and bacteremia due to *P. multocida* are rare, and more frequently observed in infants and infected adults over 60 years old [48,49]. Immunocompromised status, cancer, cirrhosis, diabetes, and chronic obstructive lung disease have been specified as predisposing factors to pasteurellosis. The most common cause of meningitis caused by *P. multocida*, estimated at almost 90%, is direct contact with an infected animal [50]. Bacterial meningitis is caused by a penetrating bite, skull fracture or skull surgery, and blood-borne bacteremia or a spread of the infection from the surrounding area. It may also be caused by directly licking, sniffing, kissing, or touching the animal [31,51,52,53].

Few interesting reports on cardiovascular infections are available in the literature. *P. multocida* bacteremia causing aortic valve dysfunction with endocarditis [54,55,56], myopericarditis following a dog bite [57], pericardial tamponade after a cat bite [58], and endocarditis in a patient with intravenous drug abuse were described [59]. Single cases of femoral artery aneurysm with large hematoma caused by *P. multocida* [60] and infection resulting in a descending thoracic aorta mycotic pseudoaneurysm were described [61]. Positive blood culture results were obtained in each reported case, and direct contact with a domestic animal was confirmed.

There are relatively many reports on *P. multocida* septicemia in the available scientific sources. The direct causes in the described cases were bites and scratches caused by dogs or cats. In most of the described cases, there were chronic comorbidities significantly worsening the prognosis. These included diabetes, cirrhosis, renal insufficiency, gout arthritis, Cushing’s syndrome, hypertension, dyslipidemia, hypothyroidism, heart failure, chronic kidney disease, immunodeficiencies, obstructive sleep apnea, and morbid obesity. An unfavorable factor in the course of the disease was the older age of the patients; some of them died [22,62,63,64,65,66,67,68,69].

A significant but rare clinical problem is peritonitis caused by *P. multocida* infection in patients undergoing peritoneal dialysis [70,71,72]. There were reports of urinary tract infection caused by direct contact with dogs and cats [73,74] and *P. multocida* infection in solid organ transplantation [75]. There were few cases of cholecystitis with positive blood and bile cultures for *P. multocida* [76], endophthalmitis in the patient with negative history of animal bites or scratch wounds [77], also ulceration on the penis caused by the teeth of a dog that was forced by the man to lick his penis have been described too [78]. A unique case of dental follicle infection following a dog bite [79] and a case of endometritis due to *P. multocida* in a woman using an intrauterine device and who has everyday contact with cats and dogs were presented in the scientific literature [80].

## 5. Differential Diagnosis

*P. multocida* is the most common pathogen isolated from wounds following animal bites or scratches. Other microorganisms such as *Bartonella henselae*, *Clostridium tetani*, *Staphylococcus aureus,* and *Rabies lyssavirus* should also be taken into account in the differential diagnosis [81].

## 6. Management

Any wound resulting from an animal bite should be disinfected. Surgical debridement, removal of debris and, in older wounds, removal of abscesses and dead tissue, and prompt use of antibiotics are essential in treating the infection. If the wound is on a limb, it is recommended to position the limb higher to minimize swelling [82,83]. The preferred and highly effective treatment against local *P. multocida* infection, if cultures cannot be acquired or *Pasteurella* is not isolated in a culture, is penicillin or its derivatives, such as oral amoxicillin-clavulanate. Rare cases of penicillin-resistant strains of *P. multocida* have been reported in human infections, and β-lactamase positivity was found in 16 percent of infected individuals [84]. Alternative therapy includes any combination of an antibiotic with anti-*Pasteurella* activity, such as doxycycline, trimethoprim/sulfamethoxazole, penicillin V, cefuroxime, ciprofloxacin, or levofloxacin, as well as an anti-anaerobic agent (metronidazole or clindamycin) to cover other oral flora. The systemic infection of *P. multocida* with positive cultures of blood, deep tissues, or respiratory tract yields the need for aggressive antibiotic treatment. The first line of parenteral antibiotic treatment includes monotherapy with ampicillin-sulbactam, piperacillin-tazobactam, or carbapenem (imipenem-cilastatin, meropenem, ertapenem). Ceftriaxone or fluoroquinolone plus an anti-anaerobic agent (such as metronidazole or clindamycin) are also acceptable. Treatment failures have been observed in patients treated with oral erythromycin, semi-synthetic penicillins such as oxacillin, dicloxacillin, as well as first-generation cephalosporins, which include cephalothin, cephalexin and cefadroxil, and clindamycin. These antibiotics have been shown to have poor in vitro activity against *P. multocida* and are, therefore, not recommended. Resistance to penicillin G, ampicillin, and tetracycline has been observed in cases of chronic pulmonary infection [85]. Antibiotic regimens should always be targeted based on cultures and sensitivities when appropriate. Treatment is usually continued for 5–14 days. The duration of antibiotic treatment should be longer in the case of a slow response or particularly severe disease [81,83].

## 7. Prophylactics

After being bitten or scratched by an animal, disinfecting and dressing the wound is essential. The primary wound closure should be carried out only if necessary, as this increases the risk of developing an infection, and a space should be left between sutures to allow drainage. Prophylactic antibiotics are not currently routinely recommended. As part of the prevention of pasteurellosis, antibiotic therapy is recommended in immunosuppressed or immunocompromised (i.e., diabetes mellitus, asplenia, cirrhosis, etc.). Amoxicillin-clavulanate is the first-line antibiotic for prophylaxis. Alternative medication for pasteurellosis prophylactics may include any antibiotic, such as doxycycline, trimethoprim/sulfamethoxazole, penicillin V, cefuroxime, ciprofloxacin, or levofloxacin, as well as an anti-anaerobic agent (metronidazole or clindamycin) to cover other oral flora [83,86].

## 8. Prognosis and One Health Prospective

The prognosis for *P. multocida* infections is usually successful, but the course of the disease depends on the infected site and the patient’s comorbidities. Most soft tissue infections resolve with the appropriate oral antibiotics and wound drainage when indicated. However, in more severe cases, such as bacteremia, meningitis, and endocarditis, the prognosis is much worse, and the mortality rate is higher (30%) [8].

## 9. Conclusions

*P. multocida* infections may affect any organ and system. This infection should be suspected especially in elderly patients with chronic diseases and in frequent contact with domestic animals, primarily cats and dogs. Avoiding bites, scratches, and direct contact with animal saliva is recommended to prevent pasteurellosis. Careful history-taking and microbiological confirmation of the infection enable diagnosis and appropriate treatment. The observed economic and social development as well as the growing material and spiritual needs of humans may be the reason for the increase in the number of domestic animals, such as cats and dogs, and farm animals used for food production. This is why *P. multocida* may become a factor threatening human health that should not be ignored.

## Data Availability

No new data were created or analyzed in this study. Data sharing is not applicable to this article.

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
