# Peer review of "Pasteurella Multocida Infection in Humans"

_pathogens, 2023, doi:10.3390/pathogens12101210_

Round 1

Reviewer 1 Report

Please see the attached PDF document.

Reviewer 2 Report

The manuscript „Pasteurella multocida Infection in Humans” summarize data about transmission, prevalence, and the clinical outcome of pasteurellosis in humans.

The manuscript „Pasteurella multocida Infection in Humans” is primarily focused on mode of transmission of P. multocida from animals to humans and the clinical signs of disease in humans. Epidemiology and clinical outcomes of pasteurelosis are described in detail, citing the relevant literature. Although some facts about pathogenesis, immunity, current diagnostic procedures, and preventive measures are missing. Thereby to improve the originality and contribution to the field, the manuscript should be completed with the missing facts and relevant data from the current literature.

There are few observations:

 1. Line 59_The sentence“ Four to five million… is twice repeated

2. Line 65_ It is not clear what percentage falls on dog and which cat bite infection 

3. Line 67_ Concerning the route of infection without direct human-animal contact_ „skin abrasion with dog saliva“ Does it means licking of the skin abrasions?

4. Authors suggest that Pasteurella infections in humans usually occur upon a bite or scratch wounds from pets. Are there any available data about transmission of Pasteurella from farm or wild animals?

5. Pasteurella multocida should be written in italic

Round 2

Reviewer 1 Report

The authors have addressed the comments, and the article is now being considered for publication.